# Reliability of Bi-Axial Ankle Stiffness Measurement in Older Adults

**DOI:** 10.3390/s21041162

**Published:** 2021-02-07

**Authors:** Hogene Kim, Sangwoo Cho, Hwiyoung Lee

**Affiliations:** 1Department of Clinical Rehabilitation Research, National Rehabilitation Center, Seoul 01022, Korea; 2Translational Research Center on Rehabilitation Robots, National Rehabilitation Center, Seoul 01022, Korea; sangwoo.cho83@gmail.com (S.C.); hwiyoung@korea.kr (H.L.)

**Keywords:** ankle stiffness, subtalar, talocrural, recliability, range of motion

## Abstract

This study involves measurements of bi-axial ankle stiffness in older adults, where the ankle joint is passively moved along the talocrural and subtalar joints using a custom ankle movement trainer. A total of 15 elderly individuals participated in test–retest reliability measurements of bi-axial ankle stiffness at exactly one-week intervals for validation of the angular displacement in the device. The ankle’s range of motion was also compared, along with its stiffness. The kinematic measurements significantly corresponded to results from a marker-based motion capture system (dorsi-/plantar flexion: *r* = 0.996; inversion/eversion: *r* = 0.985). Bi-axial ankle stiffness measurements showed significant intra-class correlations (ICCs) between the two visits for all ankle movements at slower (2.14°/s, ICC = 0.712) and faster (9.77°/s, ICC = 0.879) speeds. Stiffness measurements along the talocrural joint were thus shown to have significant negative correlation with active ankle range of motion (*r* = −0.631, *p* = 0.012). The ankle movement trainer, based on anatomical characteristics, was thus used to demonstrate valid and reliable bi-axial ankle stiffness measurements for movements along the talocrural and subtalar joint axes. Reliable measurements of ankle stiffness may help clinicians and researchers when designing and fabricating ankle-foot orthosis for people with upper-motor neuron disorders, such as stroke.

## 1. Introduction

Modern lower-limb rehabilitation devices have been refined to evaluate joint stiffness. Lokomat (Hocoma Inc., Zurich, Switzerland), for example, has been successful in clinics for lower-limb rehabilitation in patients with neurological disorders [1]. Studies have shown that the L-Stiff module in Lokomat reliably evaluated lower-limb stiffness in patients with stroke and cerebral palsy by measuring resistive torque during the flexion and extension of hip and knee joints [2,3]. Nevertheless, an unavoidable characteristic of Lokomat is the absence of an ankle unit, which causes limitations in ankle stiffness measurement. Recently, robotic devices for ankle-specific movement therapies have been developed to treat patients with neuropathological disorders [4,5,6]. The Anklebot (Interactive Motion Technology, Boston, MA, USA) intervention improved gait performance on normal surfaces in stroke patients by increasing the controllability of the affected ankle [7]. The Rutgers Ankle (Rutgers University, Piscataway, NJ, USA), a Stewart platform-type haptic device interfaced with visual feedback, was effective in gait performance after intervention with two-dimensional ankle movements in sagittal and frontal planes [8]. During assessment or training sessions, however, therapeutic ankle movements during intervention were limited to a device-dependent single degree-of-freedom. This confinement may prevent evaluation of the functional ankle stiffness in clinical trials with respect to bi-axial ankle movements. These studies may be unable to demonstrate acceptable reliability of bi-axial ankle stiffness, since the previous systems do not reflect the functional characteristics of anatomical movements of the ankle joint structure.

The ankle joint structure is intrinsically complex with relation to functional improvements to movement in elderly populations [9,10,11]. To promote efficient communication between researchers, the International Society of Biomechanics Committee recommended a three-axis ankle joint coordinate system: (1) the talocrural (or ankle) joint, which defines dorsiflexion and plantar flexion; (2) the subtalar (or talocalcaneal) joint, which defines inversion and eversion; and (3) the inferior tibiofibular joint, which defines internal and external rotations [12]. Despite the ankle joint’s structural complexity, only two main functional axes were identified: the subtalar and talocrural axes [9]. The ankle joint structure is essentially moved by four major muscle groups with diverse origins and insertions (i.e., the peroneus, tibialis, gastrocnemius, and soleus). These muscles support movements along the talocrural and subtalar joints and play an important role in the generation of stiffness in the joints during balance and gait performances [13,14]. When moving naturally, these muscles are fundamentally unable to produce perpendicular foot movements as defined. In particular, owing to its ambiguity in location and shape, the subtalar joint axis is not well defined, but has an obliquely oriented axis inclined at approximately 42° anteriorly upward from the transverse plane [9,10,11]. Thus, the anatomical characteristics of the ankle joint structure further influence the consideration of better training for the intended ankle movement functions.

Reliable ankle stiffness evaluations are therefore important to assessing the outcomes of ankle rehabilitation programs for elderly patients with neuro-musculoskeletal disorders. A custom automatic bi-axial ankle movement trainer (AMT) was developed to provide passive ankle stretching that was shown to be effective for hemiparetic gait [15]. The AMT has static and dynamic ankle stiffness measurement features, wherein the ankle torque can be measured using the anteroposterior and mediolateral vertical reaction forces while remaining still or moving bi-axially along the oblique axes with subtalar anatomical characteristics. Using the AMT, effective evaluation of the relationship between ankle stiffness and angular displacement was shown for the original mechanical property in the target ankle. A recent study demonstrated that ankle stiffness may be significantly correlated with subject-initiated active range of motion (ROM) and device-induced passive ROM [16]. 

Thus, our first objective was to evaluate the reliability of ankle stiffness measurements during ankle movement along both the subtalar and talocrural axes in elderly people, using the AMT. Then, bi-axial ankle stiffness was investigated by measuring the vertical reaction forces of the anterior, posterior, medial, and lateral areas on the moving AMT foot plate and the target foot during passive bi-axial ankle movements along the talocrural and subtalar joints, to demonstrate the test–retest reliabilities for the designated ankle movements during repeated stiffness measurements. Using these natural bi-axial ankle movements, our second aim was to investigate whether the subject-initiated active ankle ROM was correlated with ankle stiffness in the elderly. The kinematics measured with the AMT were validated with those from a conventional 3D motion-capture system. Reliable ankle static and stiffness measurements could help clinicians and orthotists provide necessary assistive devices, such as ankle-foot orthosis, to people with neurological disorders or hemiparesis. 

## 2. Materials and Methods

### 2.1. Mechanical Characteristics of the AMT System

The anatomical characteristics of the ankle joints are reflected in the current mechanical design of the AMT (Figure 1). In the movement of the talocrural joint with the AMT design, ankle dorsiflexion/plantar flexion was accomplished by the seesaw-type structure of the AMT; at a neutral ankle posture, the AMT dorsi-/plantar flexion movements were along the talocrural axis. For movement along the subtalar joint, the AMT reflected the characteristics of the subtalar joint axis, which is tilted 42° upward compared to the orthogonal ankle coordinate system. The AMT could also record inversion/eversion of the ankle while the motor was rotating, when operated with a sliding forefoot along the subtalar axis. The ankle ROM of the AMT in talocrural joint movements was limited to 50° in each direction to ensure subject safety; further, the AMT movements for the subtalar joint were limited to 12° bilaterally.

### 2.2. Design of AMT System

Two motors (Maxon Motor Inc., Sachseln, Switzerland) are installed, with one executing ankle movements along the talocrural axis, and the other executing movements along the subtalar axis. The first motor controls ankle movements along the subtalar axis, which is located at the rear of the hindfoot, 42° upward in a seesaw-type cradle (Figure 2a,b); it is connected to the footplate using a bevel gear (KGEASKG1.5-2020-10, Misumi Inc., Tokyo, Japan), because the motor rotation axis is perpendicular to the axis of the footplate. The second motor controls ankle movement along the subtalar axis, which is located in the lower part of the AMT and connected to a high-torque timing pulley along the talocrural axis via a precise timing belt.

The AMT controller unit consists of a field-programmable gate array (FPGA) with a real-time data acquisition board, integrated with a 2M gate-reconfigurable I/O FPGA board (sbRIO-9612, National Instruments, Austin, TX, USA), and two modular digital positioning controllers (EPOS2 24/2 & EPOS2 24/5, Maxon Motor Inc., Sachseln, Switzerland). The two controllers operate two DC motors via a controller area network protocol. An enclosure for these controllers and other electronic components is located under the footplate. LabView (R2015, National Instruments, Austin, TX, USA) software is used to send and receive the command data and output via ethernet protocol. To operate the AMT, two motor communication and ethernet connections were securely connected for the LabView program before beginning trials.

### 2.3. Measured Data from the AMT System

The AMT simultaneously measured underfoot reaction forces using four bar-type load cells (Shenzhen Hongrui Sensors Inc., Shenzhen, China) located under the four corners diagonally between the two rigid aluminum plates. The stiffnesses along the talocrural and subtalar axes were calculated from outputs sampled at 1000 Hz. Calibration outputs confirmed the linear characteristics of the four load cells at the four corners (mean sensor linearity: 99.99%, load < 21 kg), located between upper and lower plates, and underfoot reaction force sensors were used to measure ankle stiffness during movement along the talocrural and subtalar joints. 

Stiffness along the talocrural joint for dorsi-/plantar flexion movements was measured with two forefoot load cells, namely forefoot reaction force (FF), as well as the two hindfoot load cells, namely hindfoot reaction force (FH). Stiffness along the subtalar axis was measured with two medial-side load cells and two lateral-side load cells. The formula below defines ankle stiffness and explains its calculation along the talocrural and subtalar axes, using torques measured by vertical reaction forces from the foot plate at given ankle rotation angles. Ankle stiffness (*k_ankle_*) is defined as the slope of torques at given ankle angles with respect to the neutral ankle posture: $$k_{ankle}=\frac{\Delta T}{\Delta \theta }=\frac{T_{\theta }-T_{0}}{\theta -\theta_{0}}=\frac{\left(F_{\theta ,Fore}-F_{\theta ,Hind}\right)l_{0}-\left(F_{\theta 0,Fore}-F_{\theta 0,Hind}\right)l_{0}}{\theta -\theta_{0}}$$
where *l*_0_ is a moment arm (distance from center of force plate to load cell), $F_{\theta }$ is the vertical reaction force, Fore refers to the forefoot, Hind refers to the hindfoot, θ is a given ankle angle, and $\theta_{0}$ is a neutral ankle angle.

### 2.4. Participants and Procedures

Fifteen healthy elderly individuals without any history of neurological disorders (Female = 9, Male = 6, mean age: 76.5 years, SD: 3.1 years) were recruited for this study. All participants were briefed on the procedures and provided an informed-consent form approved by the Institutional Review Board of the National Rehabilitation Center (IRB number: NRC 2015-03-020, Seoul, South Korea). The study protocol was registered at a clinical trial registry (Clinical Research Information Service, KCT10002965). Eligibility criteria were elderly people with chronic post-stroke hemiparesis (onset duration > 6 months), who could walk independently on a level surface (Functional Ambulatory Category score > 3) without abnormal muscle tone as determined by a score on the Modified Ashworth Scale (MAS) < 3. Subjects who had complications of orthopedic disorders, cognitive dysfunctions, or mental illnesses were excluded. 

Participants performed two identical tests to assess test–retest reliability at exactly one-week intervals. A licensed physical therapist measured the active ankle ROM for the talocrural (dorsi-/plantar flexion) and subtalar joints (inversion/eversion) during the first visit. To evaluate the ankle angle measurement with the developed AMT, kinematic measurements from the AMT were synchronized with a precise 3D motion capture system using ten infrared cameras (Vicon Motion Systems Ltd., Oxford, UK) and compared to determine their correlation. To measure ankle movements, reflective markers were placed along lower-limb and trunk landmarks according to the plug-in gait marker setup [17]. Foot markers were placed on the first and fifth metatarsal joints, the calcaneus, medial and lateral malleoli, and on the rigid bar placed on midfoot (two markers). Participants were seated in a height-adjustable custom chair with their knees bent at 90°, and the dominant foot was placed on the footplate in the AMT. The AMT then produced single movements along the talocrural and subtalar joints: dorsi- and plantar flexion, as well as inversion and eversion. The participants performed passive movements of both the talocrural and subtalar joint tasks. The passive movements of talocrural joint were performed at least eight times in the range between 20° dorsiflexion and 20° plantar flexion. In addition, passive movement of the subtalar joint were performed at least eight times in the range between 12° inversion and 12° eversion. Passive ankle movements for the two tasks were both slow and fast movements. When the ankle was moving along the axes of rotation in the AMT for the talocrural and subtalar axes, the ankle kinematics, including passive ROM, were recorded at 100 Hz using a high-precision encoder with 500 counts per turn (HEDS 5540, Maxon Inc., Sachseln, Switzerland) in the AMT.

### 2.5. Statistical Analysis

Motion analysis was performed using Visual3D software (C-Motion, Inc., Germantown, MD, USA). The ankle angle was calculated using the joint angle between the dominant-side shank and dominant-side foot. Bi-axial ankle stiffness was then measured on two separate occasions in the span of one week, and test–retest reliabilities were evaluated using the SPSSWIN 18.0 software package. An intra-rater correlation, which indicates the within-subject reliability between the two visits, was assessed using a one-way intraclass correlation coefficient (ICC) random model [18]. The coefficient of variation (CV) about the relative standard error was also calculated. Inter-rater correlation was assessed with 95% confidence limits [19]. Pearson correlation was then used to assess the relationship between stiffness and active ankle ROM.

## 3. Results

Average passive ankle movement velocities were 2.14°/s (SD 0.43°/s) at the lower speed and 9.77°/s (SD 0.65°/s) at the higher speed. Passive ankle movement angle measurements using the AMT system showed considerable agreement with measurements obtained from the Vicon system (Figure 2, dorsi-/plantar flexion: mean *r* = 0.996; inversion/eversion: mean *r* = 0.985).

When the ankle attained the required position (20° at the talocrural joint and 12° at the subtalar joint) in each direction, bi-axial ankle stiffness was calculated using the four load cells. Ankle stiffness between visits 1 and 2 did not show significant differences for the different speeds (Table 1; average correlations between visits 1 and 2: slower dorsiflexion *r* = 0.683, *p* = 0.005; slower plantar flexion *r* = 0.53, *p* = 0042; slower inversion *r* = 0.69, *p* = 0.004; faster dorsiflexion *r* = 0.851, *p* < 0.001; faster plantarflexion *r* = 0.535, *p* = 0.04).

The ICC, 95% confidence interval, and limits of agreement of ankle stiffness between visits 1 and 2 are shown in Table 1. Bi-axial ankle stiffnesses showed significant reliability for slower movements, except for the stiffness around the designated everted-medial position during passive movement along the subtalar joint. 

When the ankle achieved the designated plantar flexion during slower movements, the stiffness along the talocrural joint showed a significantly negative correlation with the active ankle ROM (Figure 3, slower movement: *r* = −0.621, *p* = 0.013; faster movement: *r* = 0.6475, *p* = 0.009). Although active dorsiflexion ROM showed no significant correlation with stiffness along the talocrural joint for faster dorsiflexion movements, stiffness along the talocrural joint showed a significant positive correlation with active dorsiflexion ROM. When the ankle was in the designated eversion position during slower/faster movements, stiffness along the subtalar joint showed a significant negative correlation with active ankle ROM (slower movement: *r* = −0.521, *p* = 0.046; faster movement: *r* = −0.632, *p* = 0.011).

## 4. Discussion

To the best of our knowledge, this is the first study to reflect the complex anatomical bi-axial characteristics of the ankle joint in the design of automatic equipment for stiffness evaluation of the ankle and ankle movement treatments. Thus, it is important to reliably measure ankle stiffness during passive ankle movement. The main finding of this preliminary study is that the AMT appears to be an effective instrument to evaluate bi-axial ankle stiffness during movements along the talocrural and subtalar joints. In particular, the measurements were validated by significantly negative correlations between active ankle plantar flexion and eversion ROM and stiffness at the designated ankle positions along the talocrural and subtalar joints. Therefore, stiffness measurements using the AMT support our hypothesis that ankle stiffness during passive bi-axial ankle movement can be sufficiently evaluated with the AMT.

### 4.1. Reliability of Ankle Kinematic and Stiffness Measurements

Ankle kinematic measurements using the AMT are reliable compared to 3D kinematics using the Vicon motion capture system during passive bi-axial ankle movements (Figure 2). Our results therefore suggest that the AMT could be an effective instrument to evaluate ankle stiffness for designated ankle postures during passive movements. While the ankle joints are performing dorsiflexion, for example, the joint torque would increase more than in a neutral ankle position, causing greater stiffness along the talocrural axis in the AMT [11]. Thus, the AMT is able to evaluate posture-dependent nonlinear ankle stiffness along the full ROM of each ankle joint. Moreover, it may be possible to measure speed-dependent dynamic ankle stiffness by changing the movement speed of the AMT.

The reflection of the anatomical characteristics of the subtalar joint axis in the AMT was effective for inversion and eversion movements, which were combined movements of inversion–eversion and internal–external rotation defined within an orthogonal coordinate system. In previous studies, ankle stiffness assessment has used custom robotic ankle rehabilitation equipment, and one-degree-of-freedom ankle joint movement in terms of the orthogonal coordinate system [5,6,20]. However, the ankle structure includes three joints: the talocrural, subtalar, and inferior tibiofibular joints. In particular, the subtalar joint defines the movements between the talus and calcaneus (i.e., inversion and eversion), and the subtalar joint axis is, on average, ~42° medially oblique compared to the orthogonal coordinate system [3,9]. From our observed results (Table 1), the AMT could be an effective means of evaluating ankle stiffness during functional ankle movements along the subtalar joint, compared to measurements under orthogonal definitions. Roy et al. indicated that individuals with stroke may have significantly increased ankle stiffness during dorsiflexion and eversion [21]. In this case, our results may show that AMT measurements of ankle stiffnesses would have good reliability for stroke patients. 

The increase in disease-induced ankle joint stiffness in the elderly may cause functional decline of the lower extremities with age [22,23]. According to one study, during passive ankle dorsiflexion, the joint torque in the eldest group increased more than in any of the other younger groups [24]. Previous studies have shown that neurological diseases can cause increased joint stiffness in the geriatric population, resulting in decreased joint ROM and motoneuronal excitability [25,26], especially foot-drop syndrome in elderly patients with strokes. Foot-drop syndrome in stroke patients is affected by weakness in the ankle dorsiflexor, and ankle stiffness in stroke patients shows velocity-dependent spasticity characteristics during ankle dorsi- and plantar flexion [27]. Stroke patients clearly showed reduced ankle dorsiflexion ranges and asymmetric gait patterns between the paretic and non-paretic limbs [28]. Previous studies have also investigated methods to evaluate ankle stiffness and develop assistive devices, such as ankle–foot orthosis, to improve performance in activities of daily living in elderly and stroke patients [29,30,31]. The complexity of the ankle joint structure poses challenges to clinicians and engineers in the field of rehabilitation therapies for the development of appropriate clinical equipment specifically focused on ankle joint rehabilitation.

Bi-axial ankle stiffness was measured by using medial, lateral, forefoot, and hindfoot vertical reaction forces from the foot plate using four load cells while moving. To evaluate ankle stiffness during eversion along the subtalar joint, we decided to use the subtalar stiffness during eversion rather than that during inversion, because the custom force plate was inclined by the movement of the subtalar joint. Correlation coefficients were calculated between stiffness values of the AMT and active ankle ROMs during plantar flexion and eversion movements. Stiffness had significantly negative correlations with subject-initiated active ROM for both slower and faster ankle plantar flexion movements (Figure 3a,b). The stiffness during eversion had significantly negative correlations with the active ankle eversion ROMs (Figure 3c,d). These could be explained by the fact that older individuals with increased ankle stiffness have reduced ankle ROM [23,32]. Therefore, the AMT appears to be an effective instrument to evaluate ankle stiffness during faster movements of the subtalar joint.

### 4.2. Study Limitations

Despite the acceptable reliability of ankle stiffness measures from the test–retest evaluations, careful consideration should be taken in interpretation, because the stiffness along the subtalar axis showed less reliability at the designated eversion position during slower ankle movements (Table 1). First, full ankle ROM during passive movement of the subtalar joint is not considered in the current AMT design. Ankle ROM of the subtalar joint was designed with bilateral 24° movement; one previous study reported that the average range of ankle inversion is 0° to 35°, and the average range of ankle eversion is 0° to 15° [33]. Second, stiffness along subtalar axis may show somewhat less reliability during faster movements in the elderly. These results may be influenced by the limited ankle inversion ROM of the AMT rather than the average ROM of inversion. Extension of the ROM of subtalar joint movements is thus needed through design revision of the AMT. There may also have been a possible effect of AMT inclination approaching the maximum displacement for evaluation of ankle stiffness in the designated lateral position. Finally, data were obtained from a small sample of participants drawn from a community-based elderly population. Accordingly, further studies should include larger samples and compare the ankle stiffness characteristics of elderly individuals and stroke patients.

## 5. Conclusions

The findings from this study offer a promising step towards reliable and objective measurements of bi-axial ankle joint stiffness along the talocrural and subtalar axes. Introduction of functional subtalar movements using the AMT was successful, with high reliability in the measurement of ankle stiffness and its bi-axial kinematics. This study successfully showed a significant correlation between bi-axial ankle stiffness measurements and clinical measurements of active ankle range of motion in the elderly. It may indicate possible clinical applications of ankle stiffness measurement. With an improved design appropriate for clinical applications to ankle rehabilitation that extend the current study, AMT-based ankle stiffness measurements may enable clinical applications in relation to mobility outcomes such as gait and balance, in patients with upper-motor neuron disorders who have significant changes in ankle stiffness (i.e., stroke patients).

## 6. Patents

Patent: “Ankle Muscle Training Apparatus”, Patent No. KS10-1796916, Nov 2017, Inventors: Hogene Kim, Hwiyoung Lee, Yoon-Ho Na.

## Figures and Tables

**Figure 1 sensors-21-01162-f001:**
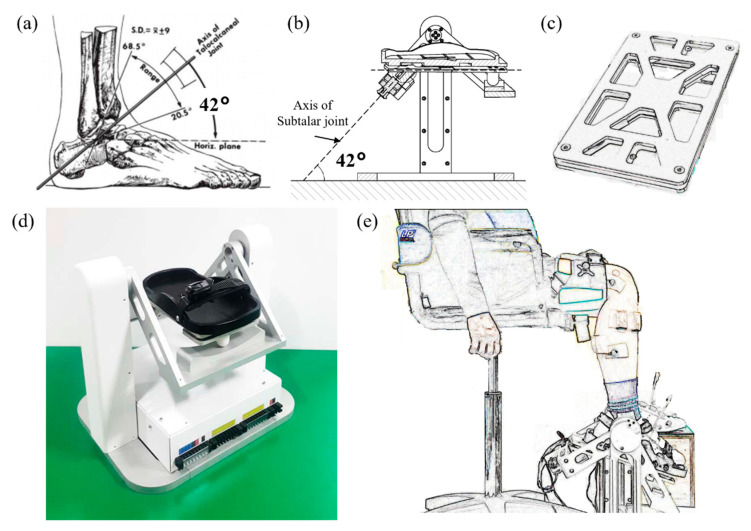
Ankle movement trainer (AMT): (**a**) Ankle anatomical characteristics (42°) in the sagittal plane of the subtalar axis [9]; (**b**) AMT mechanical design of subtalar axis; (**c**) AMT force plate with four load cells at four corners; (**d**) Ankle movement trainer (AMT); (**e**) Experimental layout of AMT human subject trial (AMT dorsi-/plantar flexion).

**Figure 2 sensors-21-01162-f002:**
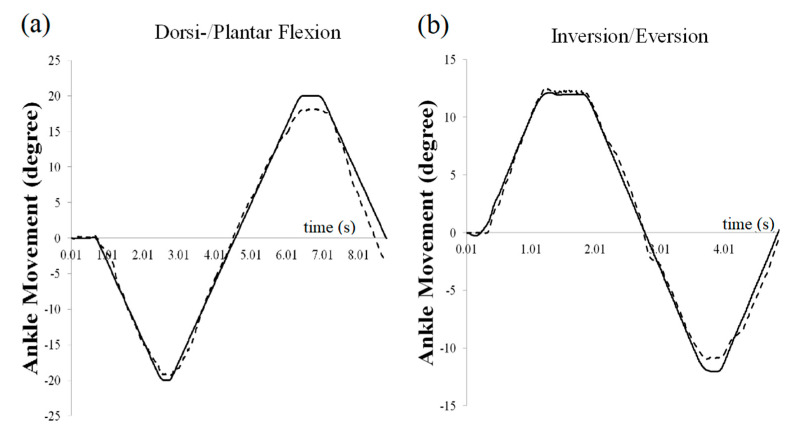
Comparison between AMT measurements and motion capture system (Vicon Motion Systems Ltd., Oxford, UK): (**a**) One cycle of dorsi-/plantar flexion movement (average *r* = 0.996, 0.983 < *r* < 0.999), (**b**) one cycle of inversion/eversion movements (average *r* =0.985, 0.959 < *r* < 0.999).

**Figure 3 sensors-21-01162-f003:**
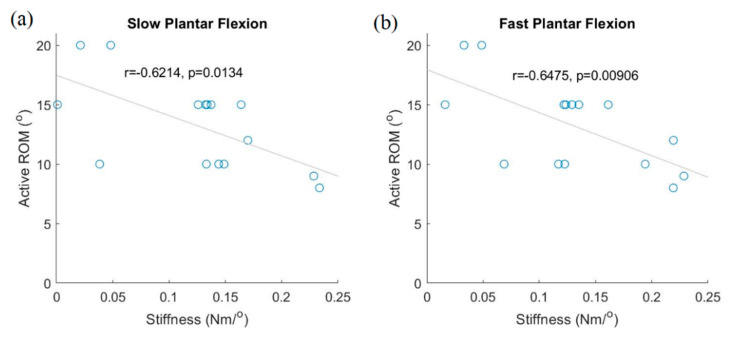
Relationship between active ankle range of motion (ROM) and ankle plantar flexion stiffness. Active plantar flexion ROM vs. ankle stiffness in plantar flexion: slower speed (**a**); faster speed (**b**). Active eversion ROM vs. ankle eversion stiffness: slower speed (**c**); faster speed (**d**).

**Table 1 sensors-21-01162-t001:** Intraclass correlation coefficients (ICC) between day 1 and day 2 on ankle stiffness for four ankle movements of 20° dorsiflexion (DF), 20° plantar flexion (PF), 12° inversion (INV), and 12° eversion (EV) using the ankle movement trainer (AMT). Ankle angular velocity was 2.14°/s for slow movement and 9.77°/s for fast movement.

		Day 1		Day 2		*r*	ICC	95% C.I.		Limits of Agreement		
		Mean	Std	Mean	Std			LB	UB	Mean	LoA-Lower	LoA-Upper
Slow Movements	DF	0.233	0.082	0.172	0.091	0.683 **	0.712 **	0.003	0.920	−0.061	−0.197	0.075
	PF	0.124	0.069	0.133	0.067	0.530 *	0.703 *	0.100	0.901	0.010	−0.120	0.139
	INV	0.067	0.073	0.028	0.076	0.690 **	0.766 **	0.266	0.923	−0.039	−0.155	0.078
	EV	−0.013	0.089	−0.051	0.090	0.419	0.575 *	−0.154	0.857	−0.038	−0.229	0.152
Fast Movements	DF	0.228	0.083	0.202	0.064	0.851 **	0.879 **	0.596	0.961	−0.026	−0.113	0.061
	PF	0.129	0.067	0.132	0.064	0.535 *	0.711 **	0.104	0.904	0.0032	0.129	−0.122
	INV	0.067	0.075	0.031	0.072	0.433	0.572 *	0.136	0.850	−0.035	−0.190	0.118
	EV	1.561	5.893	0.093	0.433	−0.071	0.282	−1.617	0.806	−1.467	−13.110	10.170

*r* denotes within-rater Pearson correlation coefficients and ICC denotes intraclass correlation coefficients between day 1 and day 2. C.I. denotes confidence interval and LB and UB denote lower and upper boundaries in the 95% confidence interval, respectively. LoA denotes limits of agreement in Bland–Altman plot. (* *p* < 0.05, ** *p* < 0.01).

## Data Availability

Not applicable.

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
