# Peer review of "Reliability of Bi-Axial Ankle Stiffness Measurement in Older Adults"

_sensors, 2021, doi:10.3390/s21041162_

Round 1

Reviewer 1 Report

The studies investigates the biaxial stiffness of an ankle in old aged people. The aim is for elderly who use a special custom angle movement trainer. The authors carry the study on 15 people with average age over 75 years old. The study evaluate the range of motion in the ankle as well as stiffness both of which are analysed. The author conclude that the method implemented can successfully measure the bi axial stiffness for reliable measurement.

The authors must avoid using we, our ..etc please check this all over the manuscript.

The authors should report of similar previous studies by other researchers if available. Show what they have done and what they found and how does your work differs from their and what contribution does this work bring in relation to others work carried out in the past.

Why specifically the angle 42 was used, also how did the authors make sure the angular dimension of the device are accurate as shown in the image 1b

how many times were the tests repeated for each of the 15 volunteers.

any obersvation regarding gender, age or previous medical history of volunteers? 

Overall the discussion is well written and explained.

However the conclusion should be explained, at least mention 3-4 bullet points on what are the most important findings from your study.

Author Response

The study investigates the biaxial stiffness of an ankle in old aged people. The aim is for elderly who use a special custom angle movement trainer. The authors carry the study on 15 people with average age over 75 years old. The study evaluated the range of motion in the ankle as well as stiffness both of which are analysed. The author conclude that the method implemented can successfully measure the bi axial stiffness for reliable measurement.

  • We appreciate the reviewer for helpful and constructive feedbacks. We have incorporated all of your suggestions into the revised manuscript as we outline in more detail in our responses below. We believe the reviewer’s critique and recommendations have greatly strengthened the manuscript.

The authors must avoid using we, our ..etc please check this all over the manuscript.

  • Thank you for the comment. We have edited it accordingly throughout the manuscript. (Lines 107,118,124,213,302,328)

The authors should report of similar previous studies by other researchers if available. Show what they have done and what they found and how does your work differs from their and what contribution does this work bring in relation to others work carried out in the past.

  • Thank you very much for your constructive comments. We have appended reports of similar previous studies and some knowledge gap, then described our contributioins in relation to others work. (Lines 43~104)
  • “Modern lower-limb rehabilitation robots have been refined to evaluate the joint stiffness . Lokomat (Hocoma Inc., Swiss), for example, has been popular in clinics for lower limb rehabilitation of patients with neurological disorders [7]. The L-Stiff module in Lokomat successfully evaluated the lower-limb stiffness in patients with stroke and cerebral palsy by measuring the resistive torque during flexion and extension of hip and knee joints[8,9]. Nevertheless, Lokomat has a fundamental limitation when measuring ankle stiffness due to the absence of ankle unit. Recently robotic devices for ankle-specific rehabilitation have been developed to treat patients with neuropathological disorders[10-12]. The Anklebot (Interactive Motion Technology, Boston, USA) intervention improved gait performance on normal surface in stroke patients by increasing the controllability of affected ankle [13]. The Rutgers Ankle (Univ. of New Jersey, USA), a Stewart platform-type haptic device interfaced with visual feedback, was effective to gait performance after intervention with 2-dimensional ankle movements in sagittal and frontal planes [14]. During assessment or training session, however, therapeutic ankle movements during intervention were limited to be a device-dependent single degree-of-freedom. This confinement may prevent to evaluate the functional ankle stiffness in clinical trials with respect to bi-axial ankle movements. These studies may be unable to show the acceptable reliability of bi-axial ankle stiffness since the measurement system has not been reflected about the functional characteristics of anatomical movements of the ankle joint structure.”

Why specifically the angle 42 was used, also how did the authors make sure the angular dimension of the device are accurate as shown in the image 1b

  • Thank you very much for the question. The angle 42 degrees specifically resembles human ankle anatomical structure. The ankle subtalar joint, one of two functional ankle joints – talocrural and subtalar(or talocalcaneal) axes, has intrinsically complicated structure that is rotating along 42 degree tilted upward axis in sagittal plane. Explanations on anatomical characteristics and AMT design were added in lines 36-42 in introduction and line 153 in methods. To explain more details about the configuration, Figure 1(a) and 1(b) were included to present the corresponding characteristics of subtalar axis.

how many times were the tests repeated for each of the 15 volunteers.

  • Thank you very much for your questions on detail descriptions about the experimental procedure. For each mono-axial movement, each subject repeated either dorsi-/plantar flexion or in-/eversion movements at least 8 times. This was described in sentences lines 236 – 240.
  • “The participants performed passive movements of both the talocrural and subtalar joint tasks. The passive movements of talocrural joint were performed at least eight times in the range between dorsiflexion of 20° and plantarflexion of 20°. In addition, the passive movements of the subtalar joint were performed at least eight times in the range between inversion of 12° and eversion of 12°. The passive ankle movement for the two tasks were slow and fast movements. When the ankle was moving along the axes of rotation in the AMT for the talocrural and subtalar axes, the ankle kinematics, including passive ROM, were recorded at 100 Hz using a high-precision encoder in the AMT.”

any observation regarding gender, age or previous medical history of volunteers? 

  • Thank you for your comments. The description on participants were appended lines 213-214 in methods section. To describe medical characteristics of volunteers, the inclusion and exclusion criteria were appended as follows and summarized in the section 2.4 Participants and Procedures.

    • Inclusion Criteria
      (1) chronic post-stroke hemiparesis,
      (2) age between 50 and 80 years,
      (3) weight less than 80 kg,
      (4) able to walk independently on level surface under supervision or sometimes using an assistive device for safety (Functional Ambulation Category (FAC) Score>3),
      (5) no abnormal muscle tone or slightly increased in ankle joint muscle group only at the end of passive ankle ROM (Modified Ashworth Scale (MAS)<3).  
    • Exclusion Criteria
      (1) complications of orthopedic disorders
      (2) cognitive dysfunctions
      (3) mental illnesses  

Overall the discussion is well written and explained. However the conclusion should be explained, at least mention 3-4 bullet points on what are the most important findings from your study.

  • Thank you very much for your constructive comments. We have updated four bulleted conclusions about the most important findings from the study in the conclusions section.

Reviewer 2 Report

  • The paper presents an interesting approach for measurements of the bi-axial ankle stiffnesses in older adults. The idea is very well described even if some other methodology should be mentioned and addresed to have a spreader view of methodology used to measure stiffness through robotic devices such as:

      Chaparro-Rico, B.D.M.; Cafolla, D.; Tortola, P.; Galardi, G. Assessing Stiffness, Joint Torque and ROM for Paretic and Non-Paretic Lower Limbs during the Subacute Phase of Stroke Using Lokomat Tools. Appl. Sci. 202010, 6168.

      Schmartz, A.C.; Meyer-Heim, A.D.; Müller, R.; Bolliger, M. Measurement of muscle stiffness using robotic assisted gait orthosis in children with cerebral palsy: A proof of concept. Disabil. Rehabil. Assist. Technol. 20116, 29–37.

  • Figure 1 miss the label description, there is a general template description please check.
  • A Layout of the experimental procedure following the description is missing, please provide it to let the reader pbetter understand what is going on during the experimental phase
  • Iclusion critia are missing 
  • Even if there is a paragraph concerning Institutional Review Board Statement and Informed Consent Statement an ethical pharagraph seems is missing describing the ethical comitee approval document and date if different from the mentioned one
  • Cocluions are very poor they should be implemented
  • Please delete Patents paragraph if you don't have patent to mention

The paper after these revision can be accepted in my opinion since it present an interesting and promising approch looking at the correlations results.

Author Response

The paper presents an interesting approach for measurements of the bi-axial ankle stiffnesses in older adults.

  • We appreciate the reviewer for helpful and constructive feedbacks. We have incorporated all of your suggestions into the revised manuscript as we outline in more detail in our responses below. We believe the reviewer’s critique and recommendations have greatly strengthened the manuscript.

The idea is very well described even if some other methodology should be mentioned and addresed to have a spreader view of methodology used to measure stiffness through robotic devices such as:

      Chaparro-Rico, B.D.M.; Cafolla, D.; Tortola, P.; Galardi, G. Assessing Stiffness, Joint Torque and ROM for Paretic and Non-Paretic Lower Limbs during the Subacute Phase of Stroke Using Lokomat Tools. Appl. Sci. 202010, 6168.

      Schmartz, A.C.; Meyer-Heim, A.D.; Müller, R.; Bolliger, M. Measurement of muscle stiffness using robotic assisted gait orthosis in children with cerebral palsy: A proof of concept. Disabil. Rehabil. Assist. Technol. 2011, 6, 29–37.

  • Thank you very much for your helpful comments. One paragraph in introduction was appended to address previous studies using robotic devices including Lokomat as mentioned in lines 43 to 104.
  • “Modern lower-limb rehabilitation robots have been refined to evaluate the joint stiffness . Lokomat (Hocoma Inc., Swiss), for example, has been popular in clinics for lower limb rehabilitation of patients with neurological disorders [7]. The L-Stiff mod-ule in Lokomat successfully evaluated the lower-limb stiffness in patients with stroke and cerebral palsy by measuring the resistive torque during flexion and extension of hip and knee joints[8,9]. Nevertheless, Lokomat has a fundamental limitation when measuring ankle stiffness due to the absence of ankle unit. Recently robotic devices for ankle-specific rehabilitation have been developed to treat patients with neuropatho-logical disorders[10-12]. The Anklebot (Interactive Motion Technology, Boston, USA) intervention improved gait performance on normal surface in stroke patients by in-creasing the controllability of affected ankle [13]. The Rutgers Ankle (Univ. of New Jersey, USA), a Stewart platform-type haptic device interfaced with visual feedback, was effective to gait performance after intervention with 2-dimensional ankle move-ments in sagittal and frontal planes [14]. During assessment or training session, how-ever, therapeutic ankle movements during intervention were limited to be a de-vice-dependent single degree-of-freedom. This confinement may prevent to evaluate the functional ankle stiffness in clinical trials with respect to bi-axial ankle movements. These studies may be unable to show the acceptable reliability of bi-axial ankle stiffness since the measurement system has not been reflected about the functional characteris-tics of anatomical movements of the ankle joint structure.”

Figure 1 miss the label description, there is a general template description please check.

  • Thank you for your comment. Appropriate legend of two more subfigures were added in Figure 1 about AMT to explain more details about AMT and experimental setup as shown in below.
  •  

A Layout of the experimental procedure following the description is missing, please provide it to let the reader better understand what is going on during the experimental phase inclusion criteria are missing 

  • Thank you for your comment. Test was done while a subject was comfortably seated at height adjustable seat with knee at 90 degrees flexed. An additional figure and details were appended in the main text to describe the test setup (lines 232-233). The subject inclusion/exclusion criteria were now updated from lines 217-223 and one figure to describe experimental setup was added in Figure 1(e).

Even if there is a paragraph concerning Institutional Review Board Statement and Informed Consent Statement an ethical paragraph seems is missing describing the ethical comitee approval document and date if different from the mentioned one.

  • Thank you very much for your thoughtful comments. Those contents were now appended in lines 214~218 as shown below.
  • “All participants were briefed on the procedures and provided the informed consent form approved by the institutional review board of the National Rehabilitation Center (IRB number: NRC 2015-03-020, Seoul, South Korea). The study protocol was registered at a clinical trial registry (Clinical Research Information Service, KCT10002965).”

Conclusions are very poor they should be implemented

  • Thank you so much for your constructive comments. Some of conclusions are now implemented as shown below (lines 2029-2039).
  • “The findings from this study offer a promising step towards a reliable and objec-tive measurements of bi-axial ankle joint stiffness along talocrural and subtalar axes. Introduction of functional subtalar movements using AMT was successful with high reliability on the measurements of ankle stiffness and its bi-axial kinematics. This study successfully showed the significant correlation between bi-axial ankle stiffness meas-urements and clinical measurements of active ankle range of motion in the elderly. It may implicate the possible clinical applications of ankle stiffness measurement. With an improved design appropriate for clinical applications to ankle rehabilitation that ex-tends the current study, AMT-based ankle stiffness measurements may enable clinical applications in relations to mobility outcomes such as gait and balance, in patients with upper motor neuron disorders, i.e. stroke patients, in the future.”

Please delete Patents paragraph if you don't have patent to mention

  • Patent information is now added in lines 2041-2042 as shown below.
  • “Patent: ‘Ankle Muscle Training Apparatus’, Patent No. KS10-1796916, Nov 2017, Inventors: Hogene Kim, Hwiyoung Lee, Yoon-Ho Na”

The paper after these revisions can be accepted in my opinion since it present an interesting and promising approach looking at the correlations results.

  • We appreciate your helpful comments on our study.

Reviewer 3 Report

The present study presents a prototype device to measure stiffness across two axes of the lower foot joint, namely the ankle and subtalar joints. Although the study is interesting, the paper fails to clearly present how it contributes to state of the art. The whole manuscript is very badly written and connection between the different sections is elusive. Introduction somewhat fails to state the gap this study aspires to fill in. Methods are badly presented, while some methodological choices are questionable. Results are not thoroughly explained, and terms like “active” and “passive” describing foot motion are confusing. Last, discussion includes over-optimistic conclusions based on the study’s results and contains parts that probably should have been in the introduction.  

Some major comments follow:

Introduction: First paragraph needs rephrasing and it should contain information on the context of the present study, for example what is the problem/knowledge gap addressed. There is some hint on the context in the last sentence of the last paragraph of the section, which should be more elaborated. Syntax errors are evident throughout the section, particularly the usage of causative sentence connectors is highly problematic.

Materials and Methods: There is no clear description or picture that describes the equipment developed to measure foot kinematics and loads. The position of the load cells is not reported.

Specific comments:

Line 102: There is no caption for Figure 1.

Line 123: At which force and axis the moment arm in the equation is referring to?

Line 128 : What are the F and M?

Lines 135-138: Have you really used one VICON camera to record the motion? This is not possible. Also, why did you use of full body marker model to record foot kinematics instead of a detailed foot marker model? This choice most probably has resulted in wrong measurements.

Line 140: How did you know which leg was dominant?

Line 147: Please, report the speed of the slow and fast movement.

Line 150: Please elaborate on this “high-precision encoder in the AMT”. How does it measure kinematics?

Results: Why do you use the term “active ROM” for a passive movement? This is very confusing.

Specific Comments:

Line 169: Average values should have been more appropriate for figure 2.

Table 1. What does LB and UB stand for? Also, the Bland-Altman method is mostly for visual comparison between two methods of measurement, hence reporting of CI and LoA values is redundant.

Discussion: This section is also very problematic and difficult to follow. Conclusions are not solid when discussing results and the logic flow is not easy to understand.

Lines 213-215: Please explain why the negative correlations between ankle eversion ROM and stiffness are a form of validation.

Lines 271: Even though you have measured healthy elderly, your conclusions on stiffness during eversion are based on stroke patients. Please explain.

Author Response

The study is interesting and well done, but the manuscript has shortcomings that you must correct:

  • We appreciate the reviewer for helpful and constructive feedbacks. We have incorporated all of your suggestions into the revised manuscript as we outline in more detail in our responses below. We believe the reviewer’s critique and recommendations have greatly strengthened the manuscript.

The title should reflect the purpose or utility of measuring Bi-axial Ankle Stiffness.

  • Thank you for your comment. Thus the title is now reflecting the purpose of this study and revised to “Reliability of Measuring Bi-axial Ankle Stiffness in Older Adults”.

You must remove the abbreviations from the Abstract.

  • We have updated the Abstract as commented to remove abbreviation AMT. Instead ICC (Intraclass Correlation Coefficients) is usually used in the abstract on any study about the test-retest reliability.

The Introduction should be expanded in general and, in particular, by referring to the research instrument used. You should develop the benefits and drawbacks of the new assessment tool that they present in comparison with the existing ones (both those for clinical application and those for application as research tools). Consequently, new bibliographic references should be used.

  • Thank you for your comments. We have appended a new paragraph (lines 43-104) that explains previous similar works with 8 new references as shown below.
  • “Modern lower-limb rehabilitation robots have been refined to evaluate the joint stiffness . Lokomat (Hocoma Inc., Swiss), for example, has been popular in clinics for lower limb rehabilitation of patients with neurological disorders [7]. The L-Stiff mod-ule in Lokomat successfully evaluated the lower-limb stiffness in patients with stroke and cerebral palsy by measuring the resistive torque during flexion and extension of hip and knee joints[8,9]. Nevertheless, Lokomat has a fundamental limitation when measuring ankle stiffness due to the absence of ankle unit. Recently robotic devices for ankle-specific rehabilitation have been developed to treat patients with neuropatho-logical disorders[10-12]. The Anklebot (Interactive Motion Technology, Boston, USA) intervention improved gait performance on normal surface in stroke patients by in-creasing the controllability of affected ankle [13]. The Rutgers Ankle (Univ. of New Jersey, USA), a Stewart platform-type haptic device interfaced with visual feedback, was effective to gait performance after intervention with 2-dimensional ankle move-ments in sagittal and frontal planes [14]. During assessment or training session, how-ever, therapeutic ankle movements during intervention were limited to be a de-vice-dependent single degree-of-freedom. This confinement may prevent to evaluate the functional ankle stiffness in clinical trials with respect to bi-axial ankle movements. These studies may be unable to show the acceptable reliability of bi-axial ankle stiffness since the measurement system has not been reflected about the functional characteris-tics of anatomical movements of the ankle joint structure.”

Zeros as the last decimal place must be removed from the entire script.

  • Thank you for your detail comments. and zeros at the last decimal place were all eliminated from the entire script.

Reviewer 4 Report

Dear Authors,

The study is interesting and well done, but the manuscript has shortcomings that you must correct:

The title should reflect the purpose or utility of measuring Bi-axial Ankle Stiffness.

You must remove the abbreviations from the Abstract.

The Introduction should be expanded in general and, in particular, by referring to the research instrument used. You should develop the benefits and drawbacks of the new assessment tool that they present in comparison with the existing ones (both those for clinical application and those for application as research tools). Consequently, new bibliographic references should be used.

Zeros as the last decimal place must be removed from the entire script.

Kind regards.

Author Response

The present study presents a prototype device to measure stiffness across two axes of the lower foot joint, namely the ankle and subtalar joints. Although the study is interesting, the paper fails to clearly present how it contributes to state of the art. The whole manuscript is very badly written and connection between the different sections is elusive. Introduction somewhat fails to state the gap this study aspires to fill in. Methods are badly presented, while some methodological choices are questionable. Results are not thoroughly explained, and terms like “active” and “passive” describing foot motion are confusing. Last, discussion includes over-optimistic conclusions based on the study’s results and contains parts that probably should have been in the introduction.  

  • We appreciate the reviewer for helpful and constructive feedbacks. We have incorporated all of your suggestions into the revised manuscript as we outline in more detail in our responses below. We believe the reviewer’s critique and recommendations have greatly strengthened the manuscript.

Some major comments follow:

Introduction: First paragraph needs rephrasing and it should contain information on the context of the present study, for example what is the problem/knowledge gap addressed. There is some hint on the context in the last sentence of the last paragraph of the section, which should be more elaborated. Syntax errors are evident throughout the section, particularly the usage of causative sentence connectors is highly problematic.

  • Thank you very much for your thoughtful comments. We have appended a paragraph reporting previous studies and knowledge gap, then described our contributions in relation to others work as follows. (Lines 43~104)
  • “Modern lower-limb rehabilitation robots have been refined to evaluate the joint stiffness . Lokomat (Hocoma Inc., Swiss), for example, has been popular in clinics for lower limb rehabilitation of patients with neurological disorders [7]. The L-Stiff module in Lokomat successfully evaluated the lower-limb stiffness in patients with stroke and cerebral palsy by measuring the resistive torque during flexion and extension of hip and knee joints[8,9]. Nevertheless, Lokomat has a fundamental limitation when measuring ankle stiffness due to the absence of ankle unit. Recently robotic devices for ankle-specific rehabilitation have been developed to treat patients with neuropathological disorders[10-12]. The Anklebot (Interactive Motion Technology, Boston, USA) intervention improved gait performance on normal surface in stroke patients by increasing the controllability of affected ankle [13]. The Rutgers Ankle (Univ. of New Jersey, USA), a Stewart platform-type haptic device interfaced with visual feedback, was effective to gait performance after intervention with 2-dimensional ankle movements in sagittal and frontal planes [14]. During assessment or training session, however, therapeutic ankle movements during intervention were limited to be a device-dependent single degree-of-freedom. This confinement may prevent to evaluate the functional ankle stiffness in clinical trials with respect to bi-axial ankle movements. These studies may be unable to show the acceptable reliability of bi-axial ankle stiffness since the measurement system has not been reflected about the functional characteristics of anatomical movements of the ankle joint structure.”

Materials and Methods: There is no clear description or picture that describes the equipment developed to measure foot kinematics and loads. The position of the load cells is not reported.

  • Thank you for your comment. Test was done while a subject was comfortably seated at height adjustable seat with knee at 90 degrees flexed. An additional figure and details were appended in the main text to describe the test setup (lines 232-233). The subject inclusion/exclusion criteria were now updated from lines 217-223. Two more Appropriate subfigures were added in Figure 1 about AMT to explain more details about AMT and experimental setup.
  • The description on participants were appended lines 213-214 in methods section. To describe medical characteristics of volunteers, the inclusion and exclusion criteria were appended as follows and summarized in the section 2.4 Participants and Procedures.

    • Inclusion Criteria
      (1) chronic post-stroke hemiparesis,
      (2) age between 50 and 80 years,
      (3) weight less than 80 kg,
      (4) able to walk independently on level surface under supervision or sometimes using an assistive device for safety (Functional Ambulation Category (FAC) Score>3),
      (5) no abnormal muscle tone or slightly increased in ankle joint muscle group only at the end of passive ankle ROM (Modified Ashworth Scale (MAS)<3).  
    • Exclusion Criteria
      (1) complications of orthopedic disorders
      (2) cognitive dysfunctions
      (3) mental illnesses  

Specific comments:

Line 102: There is no caption for Figure 1.

  • Thank you for the comment. Figure 1 is now updated with proper legend.

Line 123: At which force and axis the moment arm in the equation is referring to?

  • Thank you very much for your thoughtful comment. Appropriate comments were added in the equation (lines 210-212)

Line 128 : What are the F and M?

  • Thank you for your detail comment. It represents Female and Male. The sentence are now updated accordingly.

Lines 135-138: Have you really used one VICON camera to record the motion? This is not possible. Also, why did you use of full body marker model to record foot kinematics instead of a detailed foot marker model? This choice most probably has resulted in wrong measurements.

  • Thank you very much for your thoughtful comments one our mistakes. Our VOCON system has 10 infrared cameras and we used not full body marker model but lower limb and trunk marker setup in order to validate trunk and lower limb were properly fastened not to produce any movements during ankle movements. The foot markers are placed on the calcaneus, 1st and 5th metartarsal joints, medial and lateral malleoli and two markers were placed on the rigid bar on midfoot to measure the dorsi-/plantarflexion and in-/eversion ankle movements. Those descriptions were appended in lines 233-234.

Line 140: How did you know which leg was dominant?

  • Thank you for your comment. We have used subject reported dominance of leg usage by asking the following question. “Which leg are you kicking the ball?” This is only for the comparison between dominant and non-dominant legs for any bias on the force generation or different stiffness during passive movements. However it was negligible.

Line 147: Please, report the speed of the slow and fast movement.

  • Thank you for your comment. The slow and fast movement speed is now reported in line 275-276 as shown below.
  • “The average passive ankle movement velocities were 2.14º/s (SD 0.43º/s) at the lower speed and 9.77º/s (SD 0.65º/s) at the higher speed.”

Line 150: Please elaborate on this “high-precision encoder in the AMT”. How does it measure kinematics?

  • Thank you very much for your comment so that we could have a chance to explain details about specification of this part. The encoder (model #: HEDS 5540) is manufactured by Maxon Inc.(Swiss). This encoder is high precision encoders, DC tachometers and resolvers with a high signal resolution are mounted exclusively on motors with through shafts for resonance reasons.
  • The short description has appended in the main text (line 180).

Results: Why do you use the term “active ROM” for a passive movement? This is very confusing.

  • As reviewer commented, in many cases, we know that people are confused in clinics used the term active ROM as the ROM that human subject-initiated movements without any external helps. However, engineering people used the term “active” as actuator-initiated movements. It was a fatal error not to properly describe the setup. Thus this was updated in lines 114-116 as follows.
  • “A recent study demonstrated that ankle stiffness may be significantly correlated with subject-initiated active range of motion (ROM) and device-induced passive ROM”

Specific Comments:

Line 169: Average values should have been more appropriate for figure 2.

  • Thank you for your thoughtful comments. So the average values were now appeared in the main text instead of range.

Table 1. What does LB and UB stand for? Also, the Bland-Altman method is mostly for visual comparison between two methods of measurement, hence reporting of CI and LoA values is redundant.

  • Thank you for your comments. LB and UB stand for lower and upper boundaries of 95% confidence intervals. We The Bland-Altman plot is generally provided in this kind of studies on reliability test, we have included tables instead of the plot for efficient description because readers may appreciate more detail digits than plots about it. If necessary, we would replace it to plots. Again we appreciate your check.

Discussion: This section is also very problematic and difficult to follow. Conclusions are not solid when discussing results and the logic flow is not easy to understand.

  • Thank you very much for your comments. Thus authors tried to rewrite conclusions section to be more solid when discussing results with more logical flow so that readers could be able to follow main theme more easily.

Lines 213-215: Please explain why the negative correlations between ankle eversion ROM and stiffness are a form of validation.

  • Thank you for your question so that we could have a chance to explain more details about it. Studies have shown that there would be increased joint stiffness and reduced ROM in the elderly in general (lines 376-378). The results in this study presented outcomes with similar rationale about the relationship about less ankle ROM and greater stiffness.

Lines 271: Even though you have measured healthy elderly, your conclusions on stiffness during eversion are based on stroke patients. Please explain.

  • We appreciate your thoughtful comments. We have updated conclusions on stiffness during eversion are based on stroke patients so that the logical flow

Round 2

Reviewer 2 Report

Authors improved the paper following reviewers' suggestion.

In my opinion the paper can now be accepted for pubblication.

Author Response

We greatly appreciate all of the reviewer’s previous comments on this study.

Reviewer 3 Report

Introduction: Even if the second paragraph which was added is clearly a major improvement to the section, I believe that it should merge with the first paragraph. This is because the first paragraph should always be an introduction to the major problem this study aspires to address. Therefore, I feel that with some minor adjustments, the new second paragraph should be first.

Specific comments:

Line 143: Since lo is the distance from the center of the force plate, is the center of the joint coinciding with the center of the force plate? How did you make sure that the ankle joint center was in line with the force plate center?

This is important since you use lo to calculate torques around the ankle joint.

Author Response

Please find the attachment for authors' response on review's comments.  

Introduction: Even if the second paragraph which was added is clearly a major improvement to the section, I believe that it should merge with the first paragraph. This is because the first paragraph should always be an introduction to the major problem this study aspires to address. Therefore, I feel that with some minor adjustments, the new second paragraph should be first.

We agree the reviewer’s comment. As the reviewer commented, we revised the introduction so that the new second paragraph would be the first with some minor adjustments. The revised first paragraph in introduction now reads:

Modern lower-limb rehabilitation devices have been refined to evaluate the joint stiffness. Lokomat (Hocoma Inc., Swiss), for example, has been successful in clinics for lower limb rehabilitation of patients with neurological disorders [1]. Studies have shown that the L-Stiff module in Lokomat reliably evaluated the lower-limb stiffness in patients with stroke and cerebral palsy by measuring the resistive torque during flexion and extension of hip and knee joints[2,3]. Nevertheless, an unavoidable characteristic of Lokomat is the absence of ankle unit that would cause limitation in the ankle stiff-ness measurement. Recently robotic devices for ankle-specific movement therapies have been developed to treat patients with neuropathological disorders[4-6]. The Anklebot (Interactive Motion Technology, Boston, USA) intervention improved gait performance on normal surface in stroke patients by increasing  the controllability of affected ankle [7]. The Rutgers Ankle (Univ. of New Jersey, USA), a Stewart plat-form-type haptic device interfaced with visual feedback, was effective to gait performance after intervention with 2-dimensional ankle movements in sagittal and frontal planes [8]. During assessment or training session, however, therapeutic ankle movements during intervention were limited to be a device-dependent single degree-of-freedom. This confinement may prevent to evaluate the functional ankle stiffness in clinical trials with respect to bi-axial ankle movements. These studies may be unable to show the acceptable reliability of bi-axial ankle stiffness since the measurement system has not been reflected about the functional characteristics of anatomical movements of the ankle joint structure.

Specific comments:

Line 143: Since lo is the distance from the center of the force plate, is the center of the joint coinciding with the center of the force plate? How did you make sure that the ankle joint center was in line with the force plate center?

We appreciate the reviewer’s critique and the opportunity to clarify details about the torque measurements here. As shown in the Figure 1. the AMT foot plate is located on the seesaw-type cradle so that it makes rotational movement along an axis above the force plate. The axis of foot plate rotation is designed to be ankle(talocrural) axis so that the torque measurements from the foot force plate would be possible. As shown in the Figure below, it shows the test configuration (Left) and how the ankle stiffness was measured (right).

Subjects were asked to locate their malleolus along the AMT foot plate rotational axis, i.e. the foot plate rotation center. To make sure that the ankle joint center was in line with the force plate rotational center, there was a small LED light installed on the force plate seesaw axis whether or not the light targets coinciding on the malleolus center in testing foot. With this configuration, we could measure torques around the ankle joint using AMT.
